# Association between Changes in Daily Life Due to COVID-19 and Depressive Symptoms in South Korea

**DOI:** 10.3390/healthcare12080840

**Published:** 2024-04-16

**Authors:** Ha-Eun Son, Young-Seoub Hong, Hyunjin Son

**Affiliations:** Department of Preventive Medicine, College of Medicine, Dong-A University, Busan 49201, Republic of Korea; haundl@dau.ac.kr (H.-E.S.); yshong@dau.ac.kr (Y.-S.H.)

**Keywords:** age-related health, community health surveys, mental health, pandemics

## Abstract

We aimed to examine changes in daily life due to coronavirus disease 2019 (COVID-19) among younger (≤64 years) and older (≥65 years) individuals and to analyze their association with depressive symptoms. Raw data from the 2020 Korean Community Health Survey were used to analyze 228,485 individuals. Changes in daily life due to COVID-19 were measured using a questionnaire that evaluated changes in physical activity, sleep duration, instant food intake, and drinking and smoking status. Depressive symptoms were assessed using the Patient Health Questionnaire 9 scale, and logistic regression analysis was performed to explore the association between the two variables. This study confirmed a significant association between the two variables and found that the intake of instant food showed the largest difference in odds ratios between the younger (OR: 1.851; 95% CI: 1.720–1.992) and older groups (OR: 1.239; 95% CI: 1.060–1.447). A major finding of this study is that the analysis of the association between the two variables revealed a stronger correlation in more variables in the younger population compared to the older population. To address COVID-19-related depression and prepare for potential mental health crises, countries should expand response measures.

## 1. Introduction

In response to the global spread of the coronavirus disease (COVID-19) that was detected in December 2019, the World Health Organization (WHO) declared COVID-19 a public health emergency of international concern in January 2020 and subsequently declared it a pandemic in March 2020 [1]. The COVID-19 pandemic has not only a direct impact on physical health but also an impact on psychological health and so remains a serious threat to people worldwide [2].

As a method to control COVID-19, various countries worldwide implemented strict social distancing measures, which reduced social support and interpersonal communication, leading to public mental health crises characterized by exacerbations of depression, fear, insomnia, and anxiety symptoms [3]. Restricting social activity to prevent viral spread may hinder psychological adaptation [4]. A population’s psychological responses to an outbreak of an infectious disease not only accelerate the spread of the disease but are also major triggers of emotional distress and social disabilities even after eradication of the infectious disease [5]. The WHO and mental health experts have warned about the potential surge of mental disorders as a result of the COVID-19 pandemic and highlighted the importance of relevant research [6].

Physical and mental illnesses and emotional stress were also linked to an increased risk of experiencing depression [7]. Most studies that investigated the prevalence of depression in the general population since the outbreak of COVID-19 reported an increased prevalence of depression [8,9]. Moreover, disturbance of lifestyle, such as altered sleep and nutrition, may increase the risk of experiencing depression [10], while prolonged social distancing and shrinkage of economic activities may lead to depression [11]. COVID-19 has caused profound changes in society, and the changes in people’s daily lives caused by the pandemic have had an unprecedented impact on their mental health [2].

For this reason, the Korean Ministry of Health and Welfare implemented measures to reduce social depression since 2020, including the provision of psychological support for the public [12]. Particularly, more policies have been established and studies have been conducted targeting older adults, a population classified as vulnerable to infection. However, there is a lack of studies comparing older people with younger people who are more likely to face stress due to restrictions on daily life owing to the wider radius of daily life compared to that of older people [13].

Therefore, in the context of the pandemic, a comparative study by age is needed to compare younger and older individuals with different patterns of daily changes. Accordingly, this study aimed to identify the changes in daily life caused by the COVID-19 pandemic in younger (≤64 years) and older (≥65 years) individuals and to analyze their association with depressive symptoms.

Accordingly, this study proposes two hypotheses: First, that the changes in daily life due to the COVID-19 pandemic are associated with depressive symptoms. Second, that the relationship between changes in daily life due to the COVID-19 and depressive symptoms will differ according to age groups, suggesting that younger and older individuals may experience these associations differently.

## 2. Materials and Methods

### 2.1. Study Population and Data Collection

We used raw data from the 2020 Korean Community Health Survey (KCHS), an annual survey of adults aged ≥19 years in South Korea conducted by the Korea Disease Control and Prevention Agency (KDCA). The survey was carried out from 16 August to 31 October 2020. Of the 229,269 participants in the 2020 survey, 784 were excluded due to missing data, resulting in 228,485 participants (≤64 years: *n* = 156,150; ≥65 years: *n* = 72,335) who provided complete information about depressive symptoms and were included in the final analysis. The KCHS was approved by the Korea Centers for Disease Control and Prevention (KCDC) Institutional Review Board (IRB) (2016-10-01-P-A) on 30 June 2016. From 2017, the need for ethical approval for the KCHS was waived by the KCDC IRB as it does not fall under scope of human subject research based on the enforcement rule of the Bioethics and Safety Act. Consequently, our research involved the use of previously collected data, IRB approval was not required for this study. Elaborate details regarding the KCHS can be found elsewhere [14].

### 2.2. Independent Variables

#### 2.2.1. Sociodemographic Characteristics

The sociodemographic variables included sex, area of residence, monthly household income, education level, and marital status. Area of residence was divided into rural and urban areas. Monthly household income was divided into KRW < 3 million, KRW 3–4.9 million, and KRW ≥ 5 million (KRW 1 million = USD 749 in 2024). Education level was categorized into middle school or lower, high school, and college or higher. Marital status was categorized into living with a spouse and not living with a spouse.

#### 2.2.2. Health-Related Characteristics

The health-related characteristics included smoking status, drinking status, hypertension, diabetes mellitus, breakfast frequency, physical activity, sleep duration, and subjective stress. In terms of smoking (cigarettes), the participants were categorized as current smokers or non-smokers; in terms of drinking, the participants were categorized as drinkers and non-drinkers based on their alcohol consumption status in the previous year. Hypertension and diabetes mellitus were determined according to the physician’s diagnosis. Breakfast frequency was divided into <2 times/week and ≥2 times a week in the previous week. Physical activity was defined as the number of days per week that the participants engaged in moderate to high-intensity physical activity, defined as at least 30 min of activity that causes mild shortness of breath and physical fatigue a day for 5 days or more in the previous week. Sleep was categorized into ≥7 h and <7 h. Subjective stress was divided into high (“very stressed”; “stressed”) and low (“stressed a little”; “almost no stress”).

#### 2.2.3. Changes in Daily Life Due to the COVID-19 Pandemic

Changes in daily life due to COVID-19 were measured based on changes in the following eight factors: physical activity (e.g., walking and exercise), sleep duration, consumption of instant foods or soft drinks, use of food deliveries, drinking status, smoking status, number of days meeting friends or neighbors, and public transit. We investigated whether these activities increased, remained similar, decreased, or were not applicable compared with that during the pre-COVID-19 period. For the multivariate analysis, all occurrences of the “Not applicable” response were removed, and the remaining responses were divided into the following categories: “similar” and “changed” (increased or decreased).

### 2.3. Dependent Variables: Depressive Symptoms

Depressive symptoms were measured using the Korean version of the Patient Health Questionnaire 9 (PHQ-9) scale. The Korean version of the PHQ-9 has been validated for its reliability and validity as a screening tool for major depressive disorder [15]. The PHQ-9 is a nine-item screening tool used to measure the frequency of depression-related symptoms, including interest in work, depressive mood, sleep disorder, fatigue, appetite, concentration difficulty, anxiety behaviors, and self-depreciation. The total score ranges from 0 to 27, and a higher score indicates more severe depressive symptoms. In the present study, the participants were categorized into depressed (≥10) and non-depressed (<10) groups [16]. Cronbach’s alpha was 0.798 in this study.

### 2.4. Statistical Analysis

The participants were stratified into younger (≤64 years) and older (≥65 years) groups; then, the frequency and percentage for sociodemographic and health-related characteristics and changes in daily life due to COVID-19 in the depressed and non-depressed groups in each stratum were presented. The differences between the groups were analyzed using a chi-square test. The association between changes in daily life due to COVID-19 and depressive symptoms were analyzed using logistic regression and presented as odds ratio (OR) and confidence interval (CI). A *p*-value of 0.05 was considered significant. The analyses were performed using Stata version 17.0.

## 3. Results

### 3.1. Depressive Symptoms according to Sociodemographic and Health-Related Characteristics

The differences in sociodemographic and health-related characteristics according to the presence or absence of depressive symptoms were analyzed in younger and older groups (Table 1). Among the younger population, 2.5% (*n* = 3901) did have depressive symptoms, while 97.5% (*n* = 152,249) did not. Depressive symptoms significantly differed according to sex, area of residence, monthly household income, education level, marital status, current smoking status, current drinking status, hypertension, diabetes mellitus, breakfast frequency, moderate or higher intensity physical activity, sleep duration, and subjective stress. Among the older population, 3.5% (*n* = 2535) did have depressive symptoms, while 96.5% (*n* = 69,800) did not. The presence or absence of depressive symptoms significantly differed in all characteristics, with the exception of current smoking status (*p* = 0.804).

### 3.2. Depressive Symptoms according to Changes in Daily Life Due to COVID-19

The differences in the changes in daily life due to COVID-19 according to the presence or absence of depressive symptoms were compared between the younger and older populations (Table 2). In both populations, the depressive symptoms status significantly differed according to changes in physical activity, sleep duration, instant food intake, use of food deliveries, drinking status, smoking status, frequency of meeting friends and neighbors, and use of public transit.

### 3.3. Factors Causing Changes in Daily Life Due to the COVID-19 Pandemic Associated with Depressive Symptoms

The association between changes in daily life due to the COVID-19 pandemic and depressive symptoms in the younger and older populations was analyzed using logistic regression (Figure 1). In the adjusted analysis of the younger population, participants with changes in the following areas of daily life due to the COVID-19 pandemic were at greater odds of experiencing depressive symptoms: physical activity (OR: 1.693; 95% CI: 1.574–1.821), sleep duration (OR: 2.815; 95% CI: 2.638–3.004), instant food intake (OR: 1.851; 95% CI: 1.720–1.992), use of food deliveries (OR: 1.581; 95% CI: 1.460–1.712), drinking status (OR: 1.309; 95% CI: 1.203–1.425), smoking status (OR: 2.056; 95% CI: 1.827–2.315), and social contact (OR: 0.885; 95% CI: 0.799–0.981).

In the older population, the participants with changes in the following areas of daily life due to the COVID-19 pandemic were at greater odds of experiencing depressive symptoms: physical activity (OR: 1.573; 95% CI: 1.438–1.721), sleep duration (OR: 2.356; 95% CI: 2.142–2.590), instant food intake (OR: 1.239; 95% CI: 1.060–1.447), and social contact (OR: 0.852; 95% CI: 0.753–0.950). The association between depressive symptoms and changes in daily life due to COVID-19 was observed in both the younger and older groups, with the younger group showing higher odds ratios across multiple variables compared to the older group.

## 4. Discussion

This study analyzed the association between changes in daily life due to the COVID-19 pandemic and depressive symptoms in younger and older populations using the 2020 KCHS data.

The prevalence rates of depressive symptoms were 2.5% in the younger population and 3.5% in the older population, and changes in daily life due to the COVID-19 pandemic were associated with depressive symptoms in both groups. This finding is consistent with that of an Australian study on adults aged ≥18 years, where psychological distress, including depression, was significantly associated with changes in health behaviors [17].

One major finding of this study is that the prevalence of depressive symptoms was higher in the older group, while the association between changes in daily life due to COVID-19 and depressive symptoms was more significant in the younger group, showing more significant results across multiple variables than the older group. In the younger population, the changes in physical activity, sleep duration, instant food intake, use of food deliveries, drinking status, smoking status, and social contact were significantly associated with depressive symptoms; in the older population, the changes in physical activity, sleep duration, instant food intake, and social contact were significantly associated with depressive symptoms.

Participation in physical activity significantly alleviates depression and anxiety [18]. However, the COVID-19 pandemic can deprive people of opportunities to engage in physical activity and limit the scope of their activities. We can interpret our results in this context, in that changes in physical activity had a greater impact on younger individuals than older individuals, possibly due to their broader range of daily life activities.

In terms of sleep duration, short sleep [19] or long sleep exceeding 7–8 h [20] are linked to the onset of depression, and sleep pattern, sleep quality, and sleep satisfaction have a substantial impact on an individual’s quality of life and mental health. Hisler et al. reported that the incidence of sleep duration that is shorter or longer than the recommendation increased since the outbreak of COVID-19, and poor sleep health was more prevalent in adults aged <60 years compared with that in adults aged ≥60 years [21]. These results are consistent with our findings. Older adults have already been facing difficulties initiating and maintaining sleep compared with younger people even before the onset of COVID-19, which may explain why they are less influenced by the changes brought on by the ongoing pandemic [22]. Furthermore, in light of our result that changes in sleep duration due to COVID-19 was most strongly associated with depressive symptoms, effective measures should be applied to manage sleep quality during the COVID-19 pandemic to prevent the occurrence of serious depression.

Changes in the use of food deliveries and instant food intake during COVID-19 were also predictors of depressive symptoms, with a greater association in the younger population. Nutrition plays an important role in mental health, and Western-style diet or diet that induces inflammation, such as fast food, is linked to exacerbations of mental health problems [23,24]. During the COVID-19 pandemic, social distancing has led to increased availability of types of food that can be delivered and increased use of food deliveries; in general, popular delivery foods were foods that are less fresh than those cooked at home, such as fast foods, junk foods, coffee, and drinks [25]. The younger population was more influenced by these changes as they had easier access to these types of foods.

Smoking and alcohol consumption are also well-known risk factors for mental health problems [26,27]. According to recent studies, the prevalence of health-hazardous behaviors, such as smoking and high-risk drinking, increased during the COVID-19 pandemic in several countries [28,29]. In our study, changes in smoking and alcohol consumption due to COVID-19 were significantly associated with depressive symptoms only in the younger population. Social isolation caused by COVID-19, along with changes in employment status or uncertainty of the future, may cause individuals to consume more alcohol [30]; moreover, employment conditions are strongly associated with an individual’s smoking and alcohol drinking during the pandemic [31]. The increased prevalence of cigarette smoking and alcohol drinking among people with unstable employment status may be a strategy to cope with the mental distress caused by financial and work-related pressures.

The prolonged COVID-19 pandemic, need to maintain social distancing, and consequent changes in daily life may persist psychological distress and depressive symptoms. To prepare for the transition of COVID-19 into an endemic disease and potential outbreaks of other novel infectious diseases, governments must explore supportive measures to promote the mental health of not only older adults but also of younger generations who may be more susceptible to changes in daily life.

One key strength of this study is that we used nationally representative and reliable data of 228,485 participants in the KCHS. However, this study is not without limitations. First, the changes in daily life due to COVID-19 were measured using self-reported data; thus, the temporal relation between variables and the causal relationships of these variables were not determined due to the cross-sectional nature of the study. Additionally, the elapsed time from the onset of the pandemic to the date of the survey was not explicitly considered, which may affect the interpretation of the impact on mental health over different phases of the pandemic.

## 5. Conclusions

Changes in daily life due to the COVID-19 pandemic are associated with depressive symptoms in community-dwelling populations, with a particularly stronger association in the younger (≤64 years) population compared with that in the older (≥65 years) population. The findings of this study may present evidence for the changes in daily life due to the COVID-19 pandemic that are associated with depressive symptoms in community-dwelling individuals aged ≤64 years and those aged ≥65 years. To prepare for a potential mental health pandemic that may ensue from the COVID-19 pandemic, different countries should expand and establish response measures and strategies to deal with COVID-19-related depression; further studies are also needed to present foundational data for devising and evaluating age-specific community health plans to promote healthy living and mental health even during a pandemic in the future.

## Figures and Tables

**Figure 1 healthcare-12-00840-f001:**
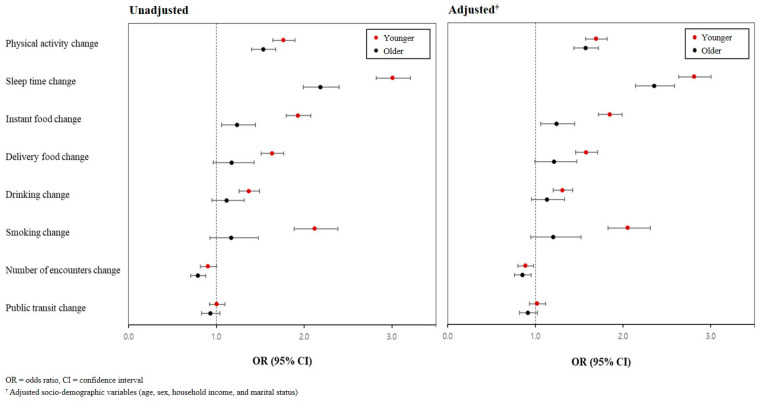
Factors of changes in daily life due to COVID-19 associated with depressive symptoms (compared to similar change in daily life).

**Table 1 healthcare-12-00840-t001:** Depressive symptoms according to the sociodemographic and health-related characteristics.

Variables	Younger (≤64 Years)						Older (≥65 Years)					
	Total	Depressed		Non-Depressed		*p*-Value	Total	Depressed		Non-Depressed		*p*-Value
	N (Column %)	N	%	N	%		N (Column %)	N	%	N	%	
Total (N = 228,485)	156,150	3901	2.5	152,249	97.5		72,335	2535	3.5	69,800	96.5	
Sex						<0.001						<0.001
Male	73,422 (47.0)	1316	1.8	72,106	98.2		30,182 (41.7)	700	2.3	29,482	97.7	
Female	82,728 (53.0)	2585	3.1	80,143	96.9		42,153 (58.3)	1835	4.4	40,318	95.7	
Residence						<0.001						<0.001
Urban	99,019 (63.4)	2742	2.8	96,277	97.2		29,678 (41.0)	1129	3.8	28,549	96.2	
Rural	57,131 (36.6)	1159	2.0	55,972	98.0		42,657 (59.0)	1406	3.3	41,251	96.7	
Household income, monthly (KRW 10,000)						<0.001						<0.001
<300	38,499 (24.7)	1552	4.0	36,947	96.0		50,466 (69.8)	2021	4.0	48,445	96.0	
300 to <500	37,173 (23.8)	797	2.1	36,376	97.9		7462 (10.3)	179	2.4	7283	97.6	
≥500	80,478 (51.5)	1552	1.9	78,926	98.1		14,407 (19.9)	335	2.3	14,072	97.7	
Educational level						<0.001						<0.001
≤Middle school	21,360 (13.7)	676	3.2	20,684	96.8		54,344 (75.1)	2173	4.0	52,171	96.0	
High school	54,240 (34.7)	1475	2.7	52,765	97.3		12,165 (16.8)	275	2.3	11,890	97.7	
≥College	80,383 (51.5)	1747	2.2	78,636	97.8		5731 (7.9)	85	1.5	5646	98.5	
Marital status						<0.001						<0.001
Living with spouse	98,115 (62.8)	1777	1.8	96,338	98.2		44,949 (62.1)	1125	2.5	43,824	97.5	
Living without spouse	57,943 (37.1)	2119	3.7	55,824	96.3		27,358 (37.8)	1409	5.2	25,949	94.9	
Smoking (current)						<0.001						0.804
No	124,933 (80.0)	2855	2.3	122,078	97.7		66,233 (91.6)	2318	3.5	63,915	96.5	
Yes	31,201 (20.0)	1046	3.4	30,155	96.7		6094 (8.4)	217	3.6	5877	96.4	
Drinking (current)						0.049						<0.001
No	47,836 (30.6)	1251	2.6	46,585	97.4		48,558 (67.1)	2010	4.1	46,548	95.9	
Yes	108,302 (69.4)	2650	2.5	105,652	97.6		23,772 (32.9)	525	2.2	23,247	97.8	
Hypertension						<0.001						<0.001
No	131,430 (84.2)	3186	2.4	128,244	97.6		33,337 (46.1)	1031	3.1	32,306	96.9	
Yes	24,710 (15.8)	714	2.9	23,996	97.1		38,985 (53.9)	1503	3.9	37,482	96.1	
Diabetes Mellitus						<0.001						<0.001
No	145,444 (93.1)	3490	2.4	141,954	97.6		56,325 (77.9)	1825	3.2	54,500	96.8	
Yes	10,697 (6.9)	411	3.8	10,286	96.2		15,998 (22.1)	708	4.4	15,290	95.6	
Eating breakfast						<0.001						<0.001
>2 times/week	53,730 (34.4)	1977	3.7	51,753	96.3		2568 (3.6)	216	8.4	2352	91.6	
≤2 times/week	102,416 (65.6)	1923	1.9	100,493	98.1		69,767 (96.4)	2319	3.3	67,448	96.7	
Moderate to vigorous physical activity						<0.001						<0.001
≥150 min/week	47,809 (30.6)	986	2.1	46,823	97.9		15,314 (21.2)	265	1.7	15,049	98.3	
<150 min/week	108,278 (69.3)	2914	2.7	105,364	97.3		56,933 (78.7)	2267	4.0	54,666	96.0	
Sleep time (weekdays)						<0.001						<0.001
≥7 h	90,686 (58.1)	1629	1.8	89,057	98.2		38,900 (53.8)	875	2.3	38,025	97.8	
<7 h	65,464 (41.9)	2272	3.5	63,192	96.5		33,435 (46.2)	1660	5.0	31,775	95.0	
Perceived stress						<0.001						<0.001
Low	115,756 (74.1)	734	0.6	115,022	99.4		62,144 (85.9)	1055	1.7	61,089	98.3	
High	40,386 (25.9)	3166	7.8	37,220	92.2		10,161 (14.0)	1475	14.5	8686	85.5	

**Table 2 healthcare-12-00840-t002:** Depressive symptoms according to changes in daily life due to COVID-19.

Variables	Younger (≤64 Years)						Older (≥65 Years)					
	Total	Depressed		Non-Depressed		*p*-Value	Total	Depressed		Non-Depressed		*p*-Value
	N (Column %)	N	%	N	%		N (Column %)	N	%	N	%	
Total (N = 228,485)	156,150	3901	2.5	152,249	97.5		72,335	2535	3.5	69,800	96.5	
Physical activity						<0.001						<0.001
Increased	8961 (5.7)	210	2.3	8751	97.7		2762 (3.8)	57	2.1	2705	97.9	
Similar	63,331 (40.6)	1086	1.7	62,245	98.3		38,829 (53.7)	1006	2.6	37,823	97.4	
Decreased	74,056 (47.4)	2268	3.1	71,788	96.9		22,688 (31.4)	939	4.1	21,749	95.9	
Not applicable	9792 (6.3)	336	3.4	9456	96.6		8038 (11.1)	530	6.6	7508	93.4	
Sleep time						<0.001						<0.001
Increased	18,321 (11.7)	642	3.5	17,679	96.5		5030 (7.0)	200	4.0	4830	96.0	
Similar	122,930 (78.7)	2185	1.8	120,745	98.2		62,795 (86.8)	1920	3.1	60,875	96.9	
Decreased	14,892 (9.5)	1073	7.2	13,819	92.8		4502 (6.2)	414	9.2	4088	90.8	
Instant food						<0.001						<0.001
Increased	26,467 (16.9)	1109	4.2	25,358	95.8		992 (1.4)	56	5.7	936	94.4	
Similar	76,362 (48.9)	1479	1.9	74,883	98.1		22,216 (30.7)	663	3.0	21,553	97.0	
Decreased	14,818 (9.5)	408	2.8	14,410	97.3		5044 (7.0)	165	3.3	4879	96.7	
Not applicable	38,469 (24.6)	904	2.4	37,565	97.7		44,046 (60.9)	1651	3.8	42,395	96.3	
Delivery food						<0.001						<0.001
Increased	45,114 (28.9)	1438	3.2	43,676	96.8		1440 (2.0)	52	3.6	1388	96.4	
Similar	52,380 (33.5)	1001	1.9	51,379	98.1		12,090 (16.7)	323	2.7	11,767	97.3	
Decreased	11,978 (7.7)	324	2.7	11,654	97.3		3430 (4.7)	100	2.9	3330	97.1	
Not applicable	46,650 (29.9)	1138	2.4	45,512	97.6		55,330 (76.5)	2059	3.7	53,271	96.3	
Drinking						<0.001						<0.001
Increased	6921 (4.4)	426	6.2	6495	93.8		594 (0.8)	30	5.1	564	95.0	
Similar	48,524 (31.1)	961	2.0	47,563	98.0		14,065 (19.4)	322	2.3	13,743	97.7	
Decreased	42,457 (27.2)	904	2.1	41,553	97.9		10,111 (14.0)	243	2.4	9868	97.6	
Not applicable	58,209 (37.3)	1608	2.8	56,601	97.2		47,528 (65.7)	1939	4.1	45,589	95.9	
Smoking						<0.001						0.003
Increased	3630 (2.3)	317	8.7	3313	91.3		334 (0.5)	16	4.8	318	95.2	
Similar	28,881 (18.5)	682	2.4	28,199	97.6		6924 (9.6)	196	2.8	6728	97.2	
Decreased	6987 (4.5)	201	2.9	6786	97.1		3160 (4.4)	99	3.1	3061	96.9	
Not applicable	116,605 (74.7)	2699	2.3	113,906	97.7		61,878 (85.5)	2223	3.6	59,655	96.4	
Number of encounters						<0.001						<0.001
Increased	479 (0.3)	26	5.4	453	94.6		228 (0.3)	9	4.0	219	96.1	
Similar	16,768 (10.7)	431	2.6	16,337	97.4		11,178 (15.5)	425	3.8	10,753	96.2	
Decreased	131,948 (84.5)	3060	2.3	128,888	97.7		55,197 (76.3)	1660	3.0	53,537	97.0	
Not applicable	6946 (4.4)	384	5.5	6562	94.5		5723 (7.9)	441	7.7	5282	92.3	
Public transit						<0.001						0.024
Increased	1149 (0.7)	69	6.0	1080	94.0		267 (0.4)	15	5.6	252	94.4	
Similar	26,606 (17.0)	776	2.9	25,830	97.1		12,014 (16.6)	457	3.8	11,557	96.2	
Decreased	45,973 (29.4)	1311	2.9	44,662	97.2		32,305 (44.7)	1140	3.5	31,165	96.5	
Not applicable	82,412 (52.8)	1745	2.1	80,667	97.9		27,741 (38.4)	922	3.3	26,819	96.7	

## Data Availability

Data are contained within the article.

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
