# Peer review of "Association between Changes in Daily Life Due to COVID-19 and Depressive Symptoms in South Korea"

_healthcare, 2024, doi:10.3390/healthcare12080840_

Round 1

Reviewer 1 Report

Comments and Suggestions for Authors

Dear authors,

Your research presented important results.

However I would like to ask you to correct the position of the chart. Somehow it separates from its caption.

The article covers a large sample of Korean citizens which is an advantage.

I would suggest the authors to give some more details on how they categorized the respondents into smokers and non-smokers as well as drinkers and non-drinkers.

Did they use WHO definitions or any other?

How did the authors determine the cutoff for the breakfast frequency and the cutoff for the sleep duration (it is known that the requirements for sleep duration differ within the ages and it might not be correct to use the same sleep duration for both age groups)?

In addition, any categorizing of numerical variables decreases the power of the statistical tests. Therefore instead of chi-square test a t-test or Mann-Whitney U test could be used for the comparison of the groups.

Reviewer 2 Report

Comments and Suggestions for Authors

Dear Authors,

The present research article, entitled “Association between Changes in Daily Life due to COVID-19 and Depression”, aims to identify the changes in daily life caused by the COVID-19 pandemic in non-elderly (≤64 years) and elderly (≥65 years) individuals and to analyze their association with depression.

The main strength of this study is that it aims to determine the association between changes in daily life due to COVID-19 and depression. This topic is very important, given the great change in life that it entailed for the entire population worldwide, especially the total social isolation of many people. Determining the consequences at the mental health level could allow us to establish specific intervention and even prevention measures according to the age of the individuals, so that we can be prepared to avoid such consequences in unexpected similar future situations.

In general, I believe that the topic and approach of this article is timely and of interest to the readers of Healthcare However, I believe that some issues should be included to improve the quality of the manuscript.

Abstract

·         It would be convenient to add something by way of conclusion and implication of the results.

·         The term "elderly" is not entirely correct because it is considered a pejorative term. I recommend using "older adults", for example.

Introduction

·         Please add the study hypothesis, which was missing in the manuscript.

Methods

·         Wasn't the variable "living alone?" taken into account? Because they may not live with their spouse but with someone else, for example, children or other relatives. The consequences may be different depending on this.

·         Were there inclusion criteria?

·         How long did the survey last?

·         During what dates did the data collection take place?

·         Was the time elapsed from the pandemic to the date of the survey taken into account? Elapsed time may be a variable affecting responses. If it was not taken into account, it can be added as a limitation of the study.

Results

·         On lines 103-105 of the methodology section, you say: " The PHQ-9 is a nine-item screening tool used to measure the frequency of depression-related symptoms, including interest in work, depressive mood, sleep disorder, fatigue, appetite, concentration difficulty, anxiety behaviors, and self-depreciation." This is true, so taking it into account, I believe that we cannot state outright that 2.5% and 3.5% of the people in the study have depression. It's a screening tool, I recommend mentioning at some point that they "probably" had depression, since no comprehensive study has been done.

Discussion

·         The image should be in the results section and not in the discussion. It should also be in the proper format (with title, etc.).

·         I recommend analyzing if there are any other limitations.

·         What possible explanation can you give for " the association between changes in daily life due to COVID-19 and depression was more significant in the non-elderly group, showing more significant results across multiple variables than the elderly group"? (Lines 167-169). By this question I mean: have the previous level of the variables been taken into account? Has there really been a change or for older people has there been no change in those variables or factors that have been taken into account?

 Best regards.

Reviewer 3 Report

Comments and Suggestions for Authors

It is valuable to analyze the relationship between changes in daily life caused by COVID-19 and depression. Due to good data analysis and manuscript writing, it is evaluated that only minor revisions are needed. Here's what seems like it needs improvement:

1. Because the changes in daily life caused by COVID-19 vary depending on the country or culture, it would be good to include the name of the nation in the title.

2. Because you used previously collected data, it may be better to mention that IRB approval was not required for this study, unlike you described.

3. Regarding the PHQ-9, it is recommended to specify the reliability and validity shown in the scale development study. In this study, it would be good to also present Cronbach's if possible.

4. The sample size is so large that it is meaningless to check the 0.05 level of significance. This is a limitation of this study. Of course, OR provides useful information.

5. Some numbers include commas after every three digits, but some do not.

Round 2

Reviewer 2 Report

Comments and Suggestions for Authors

Dear Authors,

I consider the changes you have made to be very timely.  You have taken into account the suggestions that have been proposed and have successfully addressed them.

However, I have two questions:

1.       You have responded to the recommendations made in the coverletter, why have you not included this information in the article? For example, inclusion criteria and survey duration. I recommend that this information be included.

2.       To my suggestion: What possible explanation can you give for " the association between changes in daily life due to COVID-19 and depression was more significant in the non-elderly group, showing more significant results across multiple variables than the elderly group"? (Lines 167-169). By this question I mean: have the previous level of the variables been taken into account? Has there really been a change or for older people has there been no change in those variables or factors that have been taken into account? You answer: Thank you for your question. Logistic regression analysis results (Figure 1) indicate that there was an association between depression and changes in daily life due to COVID-19 in both the Younger and Older groups. However, the odds ratios for variables showing significant associations were higher in the younger group. This is ok but could you include some possible explanation about this in the article?

Kind regards.
